# Multiparameter Persistence Images for Topological Machine Learning

**Mathieu Carrière**
DataShape
Inria Sophia-Antipolis
Biot, France
`mathieu.carriere@inria.fr`

**Andrew J. Blumberg**
Department of Mathematics
University of Texas at Austin
Austin, TX 78712
`blumberg@math.utexas.edu`

## Abstract

In the last decade, there has been increasing interest in topological data analysis, a new methodology for using geometric structures in data for inference and learning. A central theme in the area is the idea of persistence, which in its most basic form studies how measures of shape change as a scale parameter varies. There are now a number of frameworks that support statistics and machine learning in this context. However, in many applications there are several different parameters one might wish to vary: for example, scale and density. In contrast to the one-parameter setting, techniques for applying statistics and machine learning in the setting of multiparameter persistence are not well understood due to the lack of a concise representation of the results.

We introduce a new descriptor for multiparameter persistence, which we call the Multiparameter Persistence Image, that is suitable for machine learning and statistical frameworks, is robust to perturbations in the data, has finer resolution than existing descriptors based on slicing, and can be efficiently computed on data sets of realistic size. Moreover, we demonstrate its efficacy by comparing its performance to other multiparameter descriptors on several classification tasks.

## 1   Introduction

Topological data analysis (TDA) is a new and rapidly evolving branch of computer science and statistics that provides tools to analyze geometric structures in data using ideas from algebraic topology. The success of clustering methods and nonlinear dimensionality reduction techniques make it clear that even crude approaches to leveraging the geometry of data can be very effective. TDA provides more refined geometric information, and indeed, there have been a variety of successful applications of TDA, including, among others: graph analysis [CCI$^+$20, ZW19], computational biology [ACC$^+$20, CR20, GPCI15, HMMB19, KDS$^+$18, RB19], finance [dCBSB17, GGK$^+$18, GK18] and computer graphics [COO15, LOC14, PSO18].

The standard setup for the use of TDA is a data set given as a finite metric space $X$ (i.e., points and a distance function) and a continuous function $f: X \to \mathbb{R}$. This function can be understood as giving a parameter that filters $X$ and encodes how the topology changes as the parameter varies. A classical setting is when $X = \mathbb{R}^n$ with the standard Euclidean distance and $f$ is given as the distance to a point cloud $P \subset \mathbb{R}^n$, that is, $f(x) = \min \{\|x - p\| : p \in P\}$. See Figure 1. Another example is when $X$ is a graph $G$ and $f$ is a function defined on the nodes of $G$ (see Supplementary Material, Section 4).

The fundamental geometric summary of TDA, the *persistence diagram*, characterizes the changing topology of the family of sublevel sets of $f$, that is, the family $\{F_\alpha\}_{\alpha \in \mathbb{R}} = \{x \in X : f(x) \le \alpha\}_{\alpha \in \mathbb{R}}$. For instance, in the point cloud setting, the sublevel set $F_\alpha$ is the union of balls of radius $\alpha$ centered on the points of $P$. See Figure 1. Persistence diagrams are built by increasing $\alpha$ from $-\infty$ to $+\infty$,

which creates a growing sequence of sublevel sets, called a *filtration*, and recording the various topological changes that occur in this process. These changes are eventually summarized into a set of points in the plane $\mathbb{R}^2$, where each point represents a specific topological feature of the data set (e.g., a connected component, a cycle, a cavity...), and its coordinates are the parameter values $\alpha_1, \alpha_2$ for which the structure appeared and disappeared in the filtration. Often one views points away from the diagonal, i.e., which represent topological features that existed at many scales, as encoding robust geometric information and near-diagonal points as noise.

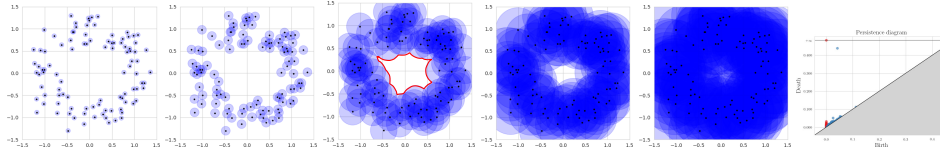

Figure 1: The filtration of the distance to the point cloud function is a sequence of union of balls with growing radii. As the radius increases, topological structures appear (e.g., the loop highlighted in red in the third union) and are eventually filled in. The "birth" and "death" values are the coordinates for a point in the persistence diagram, colored to indicate their dimension (0 is red, 1 is blue).

A problem with the summaries provided by persistence diagrams is that the space of persistence diagrams is not a convenient place to do statistics. For example, it is not a vector space, and centroids are hard to compute and not necessarily unique [TMMH14]. To handle this, a lot of work has gone into developing frameworks for supporting statistics, and machine learning for persistence diagrams has been developed in the past few years [AEK+17, BGMP14, Bub15, CCO17, CGLM15, FLR+14, KHF16, LY18, RHBK15]. However, the use of single-parameter filtrations often misses relevant information. For instance, in the point cloud setting, the scale filtration does not account for density, and so can be susceptible to outliers; for example, see Figure 4.

There are various approaches to handling the issue of variation in density in the context of the statistical frameworks for standard persistence. A more flexible general framework was introduced by Carlsson and Zomorodian: multiparameter (sometimes called multidimensional) persistence [CZ09]. This formalism encodes multiple filtration directions; the simplest example is the case of *bifiltrations*, that is, by filtering $X$ with two parameters, or functions, instead of just one. For example, point cloud outliers can be detected by simultaneously filtering by scale and density.

However, working with multiparameter persistence is substantially more difficult. Analogues of persistence diagrams can be defined in restricted settings for some specific bivariate and multivariate functions [BCB18, BLO20, CKMW20, CO19] and metrics between them have been defined, as well as algorithms for their computation [KN19, KLO19]. But in general there is no simple summary of a multiparameter persistence diagram and the integration with statistics and machine learning is still unsettled. Existing approaches for statistics in the context of bivariate functions [CFK+19, Vip20] are defined by *slicing*, that is, by considering persistence diagrams associated to linear combinations of the coordinates of the bivariate function, which is known to be limited since it is equivalent to an incomplete summary of multiparameter persistence called the rank invariant [CZ09].

In this article, we provide a more refined numerical invariant of multiparameter persistence:

- We introduce the **Multiparameter Persistence Image**, a compact descriptor which integrates information across slices by tracking changes in adjacent slices. We also investigate empirically the stability of this invariant in the face of perturbations of the slices.

- We show how to efficiently compute this descriptor using any black box matching algorithm, and provide more details for the *vineyards* algorithm [CSEM06] that we use in experiments. We also provide open-source implementation for our descriptor and for the other approaches [CFK+19, Vip20], in a public Python package [Car20].

- We demonstrate in several experiments classification experiments that the multiparameter persistence image has performance comparable to or superior to existing summaries.

## 2  Background

In this section, we briefly introduce the basics of single and multiparameter *persistent homology* as well as the *vineyard* algorithm [CSEM06] we will use to compute our descriptor. We refer the interested reader to e.g., [EH10, Car14, Oud15] for a more complete introduction to persistence.

**Single-parameter persistent homology.** Let $X$ be a topological space and $f \colon X \to \mathbb{R}$ be a continuous function. The family $\{F_\alpha\}_{\alpha \in \mathbb{R}}$ of sublevel sets of $f$, where $F_\alpha := \{x \in X : f(x) \le \alpha\}$, induces the following growing sequence of subspaces of $X$, for any parameters $\alpha_1 \le \alpha_2 \le \cdots \le \alpha_n$, called a *filtration*: $F_{\alpha_1} \subseteq F_{\alpha_2} \subseteq \cdots \subseteq F_{\alpha_n}$. Topological spaces connected by continuous maps induce vector spaces connected with linear transformations via the *homology functors* $H_k$, for $k \in \mathbb{N}$. We do not explicitly review the definition, but simply recall that each element in a basis of a vector space $H_k(F_\alpha)$ should be thought of as a $k$-dimensional *topological feature* of $F_\alpha$ (such as a connected component, a branch, a loop, a cavity, etc.), and that the linear map $v_i \colon H_*(F_{\alpha_i}) \to H_*(F_{\alpha_{i+1}})$ provides a correspondence between the features of $F_{\alpha_i}$ and $F_{\alpha_{i+1}}$. Such a sequence of vector spaces connected with linear maps is usually called the *persistence module* of $f$, denoted $M(f)$.

An essential property of persistence modules is that they can be canonically *decomposed* into a direct sum of simple modules:

$$M(f) \simeq \bigoplus_{i \in I} \mathbb{I}(\alpha_{b_i}, \alpha_{d_i}), \tag{1}$$

where $\mathbb{I}(\alpha_b, \alpha_d)$, $\alpha_b \le \alpha_d$, denotes the *interval module* between $\alpha_b$ and $\alpha_d$, which contains vector spaces of dimension 1 connected by identity maps between $\alpha_b$ and $\alpha_d$, and vector spaces of dimension 0 everywhere else. An interval module intuitively represents a topological feature of $X$ that appeared (was born) at parameter $\alpha_b$ and disappeared (died) at parameter $\alpha_d$ in the filtration.

A common way of representing decomposition (1) is to either use the values $\alpha_{b_i}$ and $\alpha_{d_i}$ as coordinates of 2D points, which leads to a set of points in the plane called the *persistence diagram* $\mathrm{dgm}(f)$, or, equivalently, as endpoints of intervals, which leads to a multiset of intervals called the *barcode* $\mathrm{bcd}(f)$. Persistence diagrams and barcodes can be equipped with a distance called the *bottleneck distance* $d_B$, which is computed by finding a partial matching which minimizes the difference between matched bars (and penalizes unmatched bars according to their length). Moreover, there is a standard distance that is commonly used to compare persistence modules, called the *interleaving distance* $d_I$. These two metrics turn out to be the same, i.e., $d_I(M(f), M(g)) = d_B(\mathrm{bcd}(f), \mathrm{bcd}(g))$. The metric lets us express the most important property of persistent homology, namely that it is *stable* with respect to this distance: $d_I(M(f), M(g)) \le \|f - g\|_\infty$, for any two continuous functions $f, g \colon X \to \mathbb{R}$ satisfying very mild hypotheses.

**Vineyards.** Persistence diagrams and barcodes are computed by constructing a filtration of a simplicial complex built from the data. The general algorithm takes $O(n^3)$ time, where $n$ is the number of simplices, which can grow rapidly in the number of data points and the range of the filtration. However, if a persistence diagram or barcode has already been computed from a function $f$, we can efficiently compute a new one for a perturbation $\tilde{f}$ of $f$ using the *vineyard* algorithm [CSEM06], which applies a sequence of $O(n)$ updates of the filtration.

Moreover, this algorithm, that we denote vine, also provides a matching between the bars of $\mathrm{bcd}(f)$ and $\mathrm{bcd}(\tilde{f})$: $\mathrm{vine}(f, \tilde{f}) = \{\mathrm{bcd}(f), \mathrm{bcd}(\tilde{f}), \mathrm{m}_{f, \tilde{f}}\}$, where $\mathrm{m}_{f, \tilde{f}}$ is a *partial matching*, that is, a 1-to-1 matching between a subset of $\mathrm{bcd}(f)$ and a subset of $\mathrm{bcd}(\tilde{f})$. In favorable cases, this matching realizes the bottleneck distance.

**Multiparameter persistent homology.** Similarly to single-parameter persistent homology, a function $f \colon X \to \mathbb{R}^d$ leads to a *multifiltration*. For any $\mathrm{a}_i, \mathrm{a}_j \in \mathbb{R}^d$, we use $\mathrm{a}_i \le \mathrm{a}_j$ to denote that each coordinate of $\mathrm{a}_i$ is smaller than the corresponding coordinate of $\mathrm{a}_j$. In this case, the sublevel sets $F_\mathrm{a} = \{x \in X : f(x) \le \mathrm{a}\}$ still satisfy $F_{\mathrm{a}_i} \subseteq F_{\mathrm{a}_j}$ as soon as $\mathrm{a}_i \le \mathrm{a}_j$, even though $\le$ is only a partial order in $\mathbb{R}^d$. For any family of parameters $\mathrm{a}_1, \ldots, \mathrm{a}_n \in \mathbb{R}^d$, the spaces $F_{\mathrm{a}_i}$ and the inclusions $F_{\mathrm{a}_i} \subseteq F_{\mathrm{a}_j}$ when $\mathrm{a}_i \le \mathrm{a}_j$ is called a *multifiltration* of $f$, which, after applying the homology functor again, gives raise to the *multiparameter persistence module $M(f)$*.

Unfortunately, general multiparameter persistence modules do not admit decompositions such as (1), except in very special cases [BCB18, BLO20, CO19]. However, for any two points $\mathrm{a}_i \le \mathrm{a}_j \in \mathbb{R}^d$, the line $\ell \colon t \mapsto (1 - t)\mathrm{a}_i + t\mathrm{a}_j$, $t \in \mathbb{R}$, defines a filtration $\{F_t\}_{t \in \mathbb{R}}$ with $F_t = \{x \in \mathbb{R}^d : f(x) \le \ell(t)\}$,

since $\ell(t_1) \leq \ell(t_2)$ for any $t_1 \leq t_2 \in \mathbb{R}$. The associated barcode, denoted by $\mathrm{bcd}(f_\ell)$ can be thought of as the barcode of the (scalar-valued) function equal to the restriction of $M(f)$ to $\ell$, and whose bars can be plotted along $\ell$ in $\mathbb{R}^d$. The collection of such barcodes is referred to as the *fibered barcode* of $f$; we use the term *slicing* to refer to the process of computing fibered barcodes. See [LW15] for an extensive discussion of the fibered barcode.

The interleaving distance of one-parameter persistence generalizes to produce a metric $d_I$ on multiparameter persistence modules, also usually referred to as the interleaving distance [Les15], which also enjoys stability properties in specific cases [BL20]. There is also a matching distance obtained as the supremum over the weighted bottleneck distances between barcodes in the fibered barcode, which serves as a lower bound for the interleaving distance.

# 3 The Multiparameter Persistence Image

Since multiparameter persistence modules cannot be readily used in statistics and machine learning applications, it is necessary to define a feature map or descriptor for it, that is, an Euclidean vector that summarizes the multiparameter persistence module. In this section we define our new descriptor, the *Multiparameter Persistence Image*. For simplicity, we restrict to the case of 2D persistence modules.

The basic idea of our construction is to select a collection of slices parametrized by lines $\{\ell_i\}$ such that there is a natural ordering on the lines, match the barcodes of adjacent lines, and consider the region in $\mathbb{R}^2$ specified by the collection of quadrilaterals with matched endpoints as vertices. In principle, one can use any black box matching algorithm, e.g., the bottleneck matching, to align the barcodes. In this work, we restrict to the vineyards algorithm for reasons of efficiency.

Let $f$ be a bivariate function $f \colon X \to \mathbb{R}^2$ and let $R = [m_1, M_1] \times [m_2, M_2] \subset \mathbb{R}^2$ be the rectangle defined by the minimum and maximum of each function coordinate: $m_1 = \min(f_1)$, $M_1 = \max(f_1)$, $m_2 = \min(f_2)$ and $M_2 = \max(f_2)$, where $f(x) = (f_1(x), f_2(x))$. We call $R$ the *bounding rectangle* of $f$. We first define the lines on which we are going to compute fibered barcodes.

**Definition 3.1.** *For any $\theta \in [0, 2\pi]$, we let $\mathbf{e}_\theta$ denote the unit vector $(\cos(\theta), \sin(\theta))$. Moreover, for any $x, v \in \mathbb{R}^2$, we let $\ell(x, v)$ denote the line passing through $x$ with direction vector $v$. Let $L_N^m = \{\ell((m_1, m_2), \mathbf{e}_{\theta_i}) : \theta_i = (i/N) \cdot (\pi/2), 0 \leq i \leq N\}$ denote $N$ ordered lines going through the lower left corner of $R$, $L_N^M = \{\ell((M_1, M_2), \mathbf{e}_{\theta_i}) : \theta_i = (i/N) \cdot (\pi/2), 0 \leq i \leq N\}$ denote $N$ ordered lines going through the upper right corner of $R$, and $L_\delta^\Delta = \{\ell((m_1 + i\delta, m_2), \mathbf{e}_{\pi/4}) : 0 \leq i \leq (M_1 - m_1)/\delta\} \cup \{\ell((m_1, m_2 + i\delta), \mathbf{e}_{\pi/4}) : 0 \leq i \leq (M_2 - m_2)/\delta\}$ denote an ordered set of evenly spaced lines with the same slope going through $R$.*

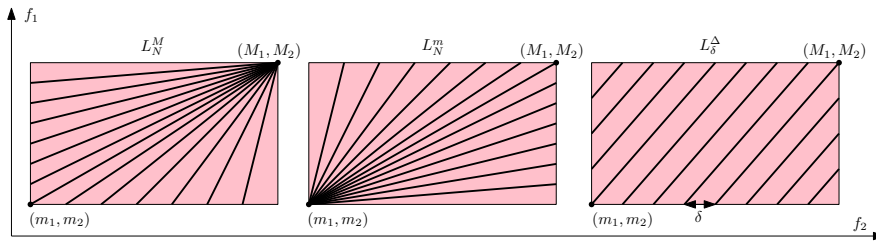

Figure 2: Examples of sets of lines used to define the Multiparameter Persistence Image.

From these sets of lines, we can use the vineyards algorithm to produce families of consecutively matched bars. This is what we call the *Vineyard Decomposition* of the multifiltration.

**Definition 3.2.** *Let $L = \{\ell_i\}_{1 \leq i \leq N}$ be one of the three sets of ordered lines of Definition 3.1.*

*The* Multiparameter Vineyard *with lines $L$, denoted by $V_L(f)$, is defined as: $V_L(f) = \mathrm{vine}(\{f_{\ell_i}\}_{1 \leq i \leq N})$, i.e., $V_L(f)$ is the list of fibered barcodes and partial matchings $\{\{\mathrm{bcd}(f_{\ell_i})\}_{1 \leq i \leq N}, \{\mathrm{m}_i\}_{1 \leq i \leq N-1}\}$, where $\mathrm{m}_i$ is a partial matching between the bars of consecutive fibered barcodes $\mathrm{bcd}(f_{\ell_i})$ and $\mathrm{bcd}(f_{\ell_{i+1}})$.*

*Equivalently, the* Vineyard Decomposition *associated to $V_L(f)$, and denoted by $D_L(f)$, is defined as the set of all subsets $I = \{b_{\ell_i}, \mathrm{m}^{(1)}(b_{\ell_i}), \mathrm{m}^{(2)}(b_{\ell_i}), \cdots\}$, where $b_{\ell_i} \in \mathrm{bcd}(f_{\ell_i})$ and $\mathrm{m}^{(p)}(b_{\ell_i}) =$*

$\mathrm{m}_{i+p} \circ \cdots \circ \mathrm{m}_i(b_{\ell_i})$ *and such that either* $i = 1$ *or* $\mathrm{m}_{i-1}^{-1}(b_{\ell_i}) = \emptyset$. *The subsets I are called the* summands *of the Vineyard Decomposition.*

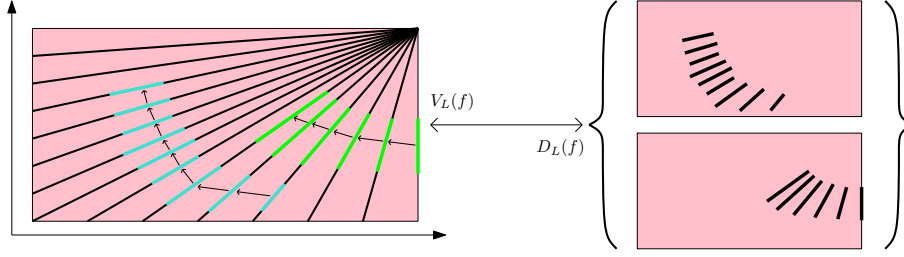

Figure 3: Example of Multiparameter Vineyard and associated Vineyard Decomposition.

The Vineyard Decomposition can be extended to multivariate functions by using multiple sets of ordered lines that always start with the same line. For example, in $R^3$, one would sweep the 3D space with planes that all intersect on a given line. Sweeping each plane with lines, 2D summands can be computed in each plane and then connected through the common line to generate 3D summands.

To produce a descriptor suitable for machine learning and statistics, we now define the *Multiparameter Persistence Image*, which is a generalization and adaptation of the *Persistence Image* for barcodes [AEK+17].

**Definition 3.3.** *Let* $R = [a, b] \times [c, d]$ *denote a rectangle in the plane* $\mathbb{R}^2$, *and* $\{P_{i,j}\}_{1 \leq i,j \leq p}$, $p \in \mathbb{N}^*$ *denote a grid of points evenly sampled on R, i.e.,* $P_{i,j} = (a + i(b - a)/p, c + j(d - c)/p)$. *The* Multiparameter Persistence Image *of resolution p and bandwidth* $\sigma > 0$ *is the matrix* $I_{L,R,p,\sigma}(f)$ *of size* $p \times p$ *such that*

$$(I_{L,R,p,\sigma}(f))_{i,j} = \sum_{I \in D_L(f)} w(I) \cdot \left( \omega(\ell^*) \exp\left( -\frac{\min_{\ell \in I} \|P_{i,j}, \ell\|^2}{\sigma^2} \right) \right), \tag{2}$$

*where* $\|P_{i,j}, \ell\| = \min_{x \in \ell} \|P_{i,j} - x\|$, $w \colon D_L(f) \to \mathbb{R}$ *is a weight function, and* $\omega(\ell^*)$ *is a weight proportional to the minimum of the entries of the vector parametrizing the line* $\ell^*$ *that achieves* $\min_{\ell \in I} \|P_{i,j}, \ell\|^2$. *Typically, we use* $w(I) = \left( \frac{A(I)}{(b-a)(d-c)} \right)^q$, *where* $A(I)$ *is the area of the convex hull of the endpoints of all bars in I and* $q \in \mathbb{N}$ *is a user-defined parameter.*

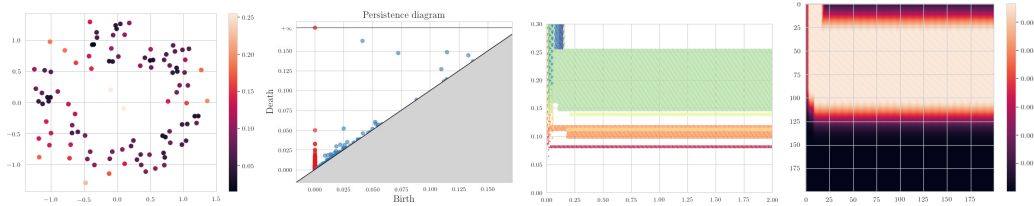

Figure 4: Single-parameter persistence fails at recovering the correct topology of a point cloud with outliers. From left to right: point cloud with outliers colored with density estimates; persistence diagram associated to the point cloud. Note the multiple blue points in dimension 1 (i.e., representing cycles) that are far away from the diagonal; Vineyard Decomposition computed with union of balls and density, the cycle is now detected as a set of long green bars; associated Multiparameter Persistence Image with resolution $p = 200$, bandwidth $\sigma = 0.001$ and power $q = 5$.

The key refinement of this construction over predecessors in the literature [CFK+19, Vip20] (see Section 4) is that we are taking the vineyards matching between bars into account when constructing the invariant, as opposed to just using the slices.

**Complexity.** The complexity associated to the Vineyard Decomposition computation is the complexity of the computation of the barcode associated to the first line $O(n^3)$ (where $n$ is the number of simplices) plus the complexity of the vineyard algorithm on the set of lines $O(N \cdot n \cdot v)$, where $N$ is the number of lines, and $v$ is the maximum number of filtration updates between consecutive

lines (and goes to 1 when $N \to +\infty$). The Multiparameter Persistence Image of resolution $p$ can be computed in $O(p^2 \cdot N \cdot m)$, where $m$ is the maximum number of bars of the fibered barcodes.

**Stability.** Work of Landi [Lan14] shows that the fibered barcode is stable in two senses. First, given a fixed line $\ell$ that has slope neither 0 nor $\infty$, parametrized by a vector $\mathbf{e}_\theta$, the assignment $M(f) \mapsto \text{bcd}(f_\ell)$ is stable in the sense that for 2D persistence modules $M(f)$ and $M(g)$, we have

$$d_B(\text{bcd}(f_\ell), \text{bcd}(g_\ell)) \le \frac{1}{\tilde{e}} d_I(M(f), M(g)) \le \frac{1}{\tilde{e}} \|f - g\|_\infty,$$

where here $\tilde{e}$ is the minimum of the entries of $\mathbf{e}_\theta$. Roughly speaking, the fibered barcode is stable in the interleaving distance with Lipschitz constant determined by the slope of the lines, where the constant approaches $\infty$ as the slope approaches either 0 or $\infty$. Second, given two lines $\ell$ and $\ell'$, Landi also shows that $\text{bcd}(f_\ell)$ and $\text{bcd}(f_{\ell'})$ are interleaved with a constant that depends on the slopes of $\ell$ and $\ell'$ and again goes to $\infty$ as the lines approach either horizontal or vertical.

The stability properties of the Multiparameter Persistence Image depend on the local stability of the matched bars in adjacent lines.

**Proposition 3.4.** *Let $\ell_1$ and $\ell_2$ be two adjacent parallel lines parameterized by the vector $\mathbf{e}_\theta$ that are distance $\delta$ apart, e.g., consecutive lines in $L_\delta^\Delta$ (see Definition 3.1). Suppose that $b_1 \in \text{bcd}(f_{\ell_1})$ and $b_2 \in \text{bcd}(f_{\ell_2})$ are matched bars in the barcodes along $\ell_1$ and $\ell_2$ respectively. Given another 2D persistence module $M(g)$ such that $d_I(M(f), M(g)) < \epsilon$, then the change in the area of the quadrilateral specified by $b_1$ and $b_2$ is bounded by $(2\delta\epsilon)/\tilde{e}$. Along a path of $k$ parallel lines at most $\delta$ apart, the change in area of the region traced out by the bars is bounded by $k \cdot (2\delta\epsilon)/\tilde{e}$.*

What the preceding proposition shows is that provided that the series of matchings given by the vineyards algorithm realize the bottleneck metric and are themselves stable under perturbation, any invariant based on the area of these regions which is weighted appropriately to compensate for the length of the path (i.e., the number of bars) and the slope of the lines will be stable to perturbation of the multiparameter persistence module and multifiltration in the interleaving distance or matching distance. We give precise theorem statements (and proofs) in Supplementary Material, Section 1.

However, the issue of the stability of the matchings itself is a subtle one. In situations where bars merge and split apart again along the vineyard, the matching can be very sensitive to infinitesimal changes in the bars. There are two approaches to handling this. One possibility is to average the images over random perturbations of the input persistence module or the matchings. However, in practice, we can algorithmically detect whether such instability can occur for a given data set, by looking at the matchings that the bottleneck distance would produce: if multiple partial matchings achieve the bottleneck distance, then choosing arbitrarily between them might make the algorithm lose track of the topological feature. Moreover, we expect this kind of bad behavior to disappear for some well-behaved classes of multifiltrations [BLO20, CO19].

## 4 Experiments

In this section, we compare the Multiparameter Persistence Image to two other descriptors that have recently been proposed for multiparameter persistence, the *Multiparameter Persistence Landscape* [Vip20] and the *Multiparameter Persistence Kernel* [CFK$^+$19]. In the following, we only describe their computational approximations for sake of simplicity. See the associated reference for their precise definitions. Let $L$ denote a set of lines (see, e.g., Definition 3.1).

**Multiparameter Persistence Landscape [Vip20].** Let $\lambda_k \colon \mathcal{B} \to \mathbb{R}^d$ denote the usual $k$th-landscape vector construction (see [Bub15] for details), which turns a barcode into an Euclidean vector obtained as a sampling of the piecewise-linear function corresponding to the $k$-th landscape. The *Multiparameter Persistence Landscape* of a bivariate function $\Lambda_k(f) \in \mathbb{R}^{d \times |L|}$ is defined as the concatenation of all landscapes associated to the fibered barcodes $\{\text{bcd}(f_\ell)\}_{\ell \in L}$ where the lines in $L$ have slope 1, which is the case when, e.g., $L = L_\delta^\Delta$ (see Definition 3.1).

**Multiparameter Persistence Kernel [CFK$^+$19].** Let $K \colon \mathcal{B} \times \mathcal{B} \to \mathbb{R}$ be any kernel between barcodes (see [CCO17, KHF16, LY18, RHBK15] for examples of possible kernels). The associated *Multiparameter Persistence Kernel* between two bivariate functions $\mathcal{K}_K(f, g)$ is defined as $\mathcal{K}_K(f, g) = \frac{1}{|L|} \sum_{\ell \in L} w(\ell) \cdot K(\text{bcd}(f_\ell), \text{bcd}(g_\ell))$, where $w(\ell)$ is a weight function that depends on the slope of $\ell$ (in order to guarantee stability).

A potential advantage of the the Multiparameter Persistence Image is that both of these invariants depend only on the fibered barcode; persistence modules with the same fibered barcodes will end up having the same Multiparameter Persistence Landscapes and Kernels, but different Multiparameter Persistence Images. See Supplementary Material, Section 2 for an example.

**Results.** All experiments have been run on an AWS machine with a Xeon Platinum 8175 processor. Code for computing the descriptors and running the experiments is freely available [Car20].

**Time series.** Our first series of experiments involve time series classification. Time series analysis, e.g., periodicity detection, can be applied to the Takens embeddings of the time series [PH15]. However, noise in the series might induced outliers in the resulting point cloud embeddings. Hence, it is natural to filter based on both distance to the point cloud and density estimates—see Figure 4 and [ACG$^+$18]. We use time series data sets from the UCR archive [DBK$^+$18] with moderate sizes and lengths; this ensures that the kernel matrices obtained with Multiparameter Persistence Kernel have reasonable sizes and that the point clouds obtained with the Takens embedding in $\mathbb{R}^3$ have a reasonable number of points. Moreover, we use the train/test split that is suggested for each data set. See Supplementary Material, Table 1 for a description of the time series classification tasks.

We used $L_\delta^\Delta$ (see Definition 3.1) as our set of lines for computing vineyards and fibered barcodes (in homological dimensions 0 and 1) for each descriptor[1], with $\delta = ((M_1 - m_1) + (M_2 - m_2))/N$, where $m_1, M_1$ are the minimum and maximum distances between all points of all point cloud embeddings, and $m_2, M_2$ are the minimum and maximum of the density estimates of all points of all point cloud embeddings, and $N = 200$. Density estimation is performed with the Distance-To-Measure [CFL$^+$18] with parameter $m = 0.1$.

We compare the accuracies of the Multiparameter Persistence Kernel computed with the Sliced Wasserstein Kernel between barcodes [CCO17] (reported as "MP-K"), the sum of the first 5 Multiparameter Persistence Landscapes (reported as "MP-L"), and the Multiparameter Persistence Image (reported as "MP-I"). Resolutions for Multiparameter Persistence Landscapes and Images are 5-fold cross-validated over the set of values $\{10, 50\}$, and the powers and bandwidths of the Multiparameter Persistence Images and Kernels are also 5-fold cross-validated with values in $\{0, 1\}$ (power) and $10^{\{-2,-1,0,1,2\}}$ (bandwidth). We also compared to accuracies obtained with 1D persistence barcodes computed along the diagonal of the bounding rectangle and vectorized with the sum of the first five 1D-persistence landscapes (reported as "P-L") [Bub15], the 1D persistence image (reported as "P-I") [AEK$^+$17], and the persistence scale-space kernel[2] (reported as "PSS-K") [RHBK15]). Resolutions, bandwidths and powers for these descriptors were cross-validated as in the multiparameter case. Finally, we add the accuracies obtained with Euclidean nearest neighbor (B1), dynamic time warping with optimized (B2) and constant window width (B3), as provided and explained in [DBK$^+$18]. Our goal in this experiment is primarily to compare different topological summaries, although it is also interesting to see the relationship to the state of the art.

All features were trained with an XGBoost classifier, except for MP-K and PSS-K, which were trained with kernel Support Vector Machines. Results are displayed in Table 1 and computation time for homological dimension 0 are in Table 2 (homological dimension 1 can be found in Supplementary Material, Table 2). The Multiparameter Persistence Image clearly outperforms the other techniques on some data sets (such as ECG200, Plane, SwedishLeaf, MedicalImages), and remain very competitive on the others. It is also clearly faster to compute, since the Multiparameter Persistence Kernel requires computing several kernel matrices, which is costly and scales poorly, and Multiparameter Persistence Landscapes requires sorting the landscape values on every sample of every slice. One can see that multiparameter persistence summaries are clearly superior to the diagonal 1D-persistence summaries. We also confirmed that the differences in accuracies were not explained only by the resolutions of the final model selected by cross-validation in Supplementary Material, Table 3, and provided more details on cross-validation results in Supplementary Material, Table 4.

A second series of experiments focused on graph classification (see Supplementary Material, Section 4). In this case, we found that all of the multiparameter persistence summaries had essentially the same performance, and our conclusion is that the fibered barcodes alone already contain all the salient topological information. Note that the fibered barcodes remain again superior to 1D-persistence.

| Dataset | B1 | B2 | B3 | PSS-K | P-I | P-L | MP-K | MP-L | MP-I |
|---|---|---|---|---|---|---|---|---|---|
| DistalPhalanxOutlineAgeGroup | 62.6 | 62.6 | **77.0** | **76.9** | 69.8 | 70.5 | 67.6 | 70.5 | 71.9 |
| DistalPhalanxOutlineCorrect | 71.7 | **72.5** | 71.7 | 47.5 | 67.4 | 66.3 | **74.6** | 69.6 | 71.7 |
| DistalPhalanxTW | **63.3** | **63.3** | 59.0 | **71.5** | 59.0 | 56.1 | 61.2 | 56.1 | 61.9 |
| ProximalPhalanxOutlineAgeGroup | 78.5 | 78.5 | **80.5** | 75.9 | **82.0** | 78.0 | 78.0 | 78.5 | 81.0 |
| ProximalPhalanxOutlineCorrect | **80.8** | 79.0 | 78.4 | 78.4 | 72.2 | 72.5 | 78.7 | 78.7 | **81.8** |
| ProximalPhalanxTW | 70.7 | **75.6** | **75.6** | 61.4 | 72.2 | 73.7 | **79.5** | 73.2 | 76.1 |
| ECG200 | **88.0** | **88.0** | 77.0 | 67.0 | 74.0 | 74.0 | 77.0 | 74.0 | **83.0** |
| ItalyPowerDemand | **95.5** | **95.5** | 95.0 | - | 64.7 | 61.1 | **80.7** | 78.6 | 79.8 |
| MedicalImages | 68.4 | **74.7** | 73.7 | 51.1 | 46.2 | 44.3 | 55.4 | 55.7 | **60.0** |
| Plane | 96.2 | **100.0** | **100.0** | 82.9 | 64.8 | 82.9 | 92.4 | 84.8 | **97.1** |
| SwedishLeaf | 78.9 | **84.6** | 79.2 | 81.0 | 37.1 | 38.2 | 78.2 | 64.6 | **83.8** |
| GunPoint | **91.3** | **91.3** | 90.7 | 90.6 | 84.7 | 80.0 | 88.7 | **94.0** | 90.7 |
| GunPointAgeSpan | 89.9 | **96.5** | 91.8 | - | 84.5 | 87.0 | **93.0** | 85.1 | 90.5 |
| GunPointMaleVersusFemale | 97.5 | 97.5 | **99.7** | - | 88.3 | 87.3 | **96.8** | 88.3 | 95.9 |
| GunPointOldVersusYoung | 95.2 | **96.5** | 83.8 | - | 98.7 | 95.9 | 99.0 | 97.1 | **100.0** |
| PowerCons | **93.3** | 92.2 | 87.8 | - | 83.4 | 76.7 | 85.6 | 84.4 | **86.7** |
| SyntheticControl | 88.0 | 98.3 | **99.3** | 50.0 | 45.7 | 44.0 | 50.7 | **60.3** | 60.0 |

Table 1: Classification results for time series.

| Dataset | MP-K | MP-L | MP-I |
|---|---|---|---|
| DistalPhalanxOutlineAgeGroup | 9227.1 | 1038.9 | **217.1** |
| DistalPhalanxOutlineCorrect | 36734.6 | 3492.6 | **833.7** |
| DistalPhalanxTW | 9396.4 | 577.7 | **138.4** |
| ProximalPhalanxOutlineAgeGroup | 11573.1 | 759.5 | **244.5** |
| ProximalPhalanxOutlineCorrect | 30822.7 | 2169.5 | **497.6** |
| ProximalPhalanxTW | 11641.7 | 375.4 | **93.4** |
| ECG200 | 1615.3 | 1355.6 | **269.0** |
| ItalyPowerDemand | 41918.1 | 1939.0 | **417.5** |
| MedicalImages | 147668.1 | 2404.7 | **599.5** |
| Plane | 2036.0 | 1065.0 | **249.2** |
| SwedishLeaf | 38045.7 | 3329.3 | **693.5** |
| GunPoint | 1977.0 | 1685.7 | **422.1** |
| GunPointAgeSpan | 14013.9 | 3945.6 | **1078.6** |
| GunPointMaleVersusFemale | 14069.9 | 4058.8 | **1097.0** |
| GunPointOldVersusYoung | 16668.1 | 5400.9 | **1388.5** |
| PowerCons | 8808.3 | 3234.8 | **811.4** |
| SyntheticControl | 13340.0 | 595.1 | **161.9** |

Table 2: Computation time (s) for time series in dimension 0.

**Immunofluorescence images.** Our second experiment deals with *quantitative immunofluorescence images*. The data is a set of pairs of images, where each pair consists of imaging data from a piece of human tissue from a patient suffering from breast cancer. The pixel intensities in an image denote the abundance of cells from one of two types: immune cells or cancer. That is, in one image cancer cells are bright, and in the other image immune cells are. These pixel intensities were obtained by injecting biomarkers in the tissue that would brighten the cells of the specified type. Moreover, the pixels of the two images are in correspondence since they represent the same location on the human piece of tissue. See [ACC+20] for more details.

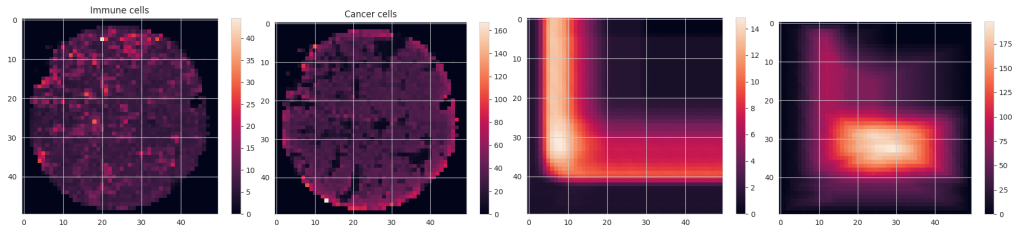

Figure 5: From left to right: immune brightness, cancer brightness, Multiparameter Persistence Image in dimension 1, Multiparameter Persistence Landscape in dimension 1. Note how the bright spots of immune cells correspond to the dark spots of cancer cells.

Analysis of these images is usually done by looking at the nearest neighbor distance distributions, that is, for each cancer pixel (i.e., pixel whose brightness is above a manual threshold in the cancer image), compute the distance to its nearest immune pixel, and vice-versa. This gives two empirical distributions, whose means and variances can be used as features for classification [GMH+18, SBSO16]. However, this approach is known to be unstable (due to the use of nearest distances), requires manual thesholding, and only represents the local neighborhoods around the cell without looking at relationships at larger spatial scales.

Since the two images are in correspondence, we can regard the brightness information as providing a pair of filter functions—this setup lends itself to analysis using multiparameter persistence. The classification task was to distinguish patients that were alive at the latest follow-up after diagnosis from those who passed away, based on the images. We used the same parameters as detailed in the previous experiments (since all 1D-persistence summaries led to the same results, we provided only one score, reported as "P"), and added a column "NN-F" for the performance of nearest neighbor distribution features fed to an XGBoost classifier. Results were averaged over 5 folds of the full 688 patients and are provided in Table 3. One can see that multiparameter persistence (and in particular the Multiparameter Persistence Image) provides a striking improvement in survival prediction over either the standard method or 1D-persistence.

| NN-F | P | MP-K | MP-L | MP-I |
|------|---|------|------|------|
| $67.2 \pm 1.0$ | $71.1 \pm 1.1$ | $77.6 \pm 2.3$ | $76.5 \pm 2.7$ | **$79.1 \pm 2.2$** |

Table 3: Classification results for immunofluorescence data.

**Stability.** We conclude with an empirical study of the stability of the Multiparameter Persistence Image with respect to slice perturbations. We perturbed the endpoints of the slices for an arbitrary time series in ECG200 with random noise of increasing amplitude, and looked at the ratio between the 1-norm of the difference between the perturbed and clean, i.e., with no noise, Multiparameter Persistence Images, and the (normalized) noise amplitude. We also checked the accuracies on ECG200. It is clear from Figure 6 that the stability ratio looks constant and even starts to decrease when the noise amplitude is large enough, clearly showing that Multiparameter Persistence Images are empirically stable. This is also observed in the accuracy, which actually increases when noise amplitude is small before slowly decreasing as well. This also indicates that optimizing over the endpoints of the lines is another potential way of increasing the performance of models.

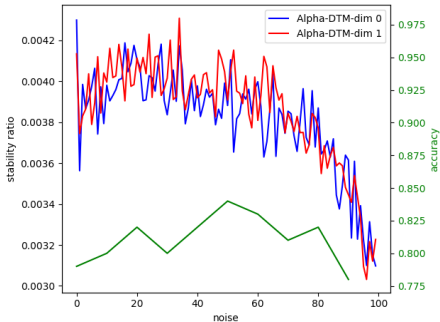

Figure 6: Empirical illustration of Multiparameter Persistence Image stability

## 5 Conclusion and future work

In this article, we introduced the *Multiparameter Persistence Image*, a compact descriptor for multiparameter persistence which is easily computable, robust to noise, and can be used for machine learning and statistics. Experiments demonstrated that its refined discriminating capability enables it to match or outperform other methods on a variety of tasks, most notably predicting survival from images of cancerous tumors. In future work, we plan to study the relation between the multiparameter persistence module decomposition into interval modules (when it exists) and the one produced by the Vineyard Decomposition. Another next step is to extend our descriptor for multiparameter persistence to larger numbers of filter functions.

## Broader Impact

Multiparameter persistence descriptors are sorely needed to enhance the reach and usability of topological data analysis, since many applications require understanding and encoding multiple filtrations at once. We believe that proposing an efficient descriptor that encodes strictly more information than just the union of all 1D-persistence diagrams associated to slices, and enabling the community to use it with an easy-to-use Python package, can have a significant impact and is an important contribution to data science and topological data analysis. In particular, we hope that our work helps to make topological methods within machine learning more accessible to practitioners.

## Acknowledgements and Funding Disclosure

Part of this work was done while the first author was working at Columbia University and funded by the United States National Institutes of Health (NIH) grant T15LM007079-28. The second author was partially funded by NIH grants 5U54CA193313 and GG010211-R01-HIV, AFOSR grant FA9550-18-1-0415, and NSF grant CNS-1514422. The authors would like to thank Mike Lesnick for helpful discussion of the algorithm and in particular the issue of Vineyard stability. The authors would also like to acknowledge Kevin Gardner's lab at Columbia University for kindly sharing their immunofluorescence images.

## Footnotes

[1]We also tried the other sets of lines in Definition 3.1 for the Multiparameter Persistence Kernel and Image, but did not report the results since changes were not significant.

[2]In a few tasks where the PSS-K values are infeasible to compute, we did not report accuracy.

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
