[Supplementary Material]

# Supplementary Material for "Multiparameter Persistence Image for Topological Machine Learning"

**Mathieu Carrière**
DataShape
Inria Sophia-Antipolis
Biot, France
`mathieu.carriere@inria.fr`

**Andrew J. Blumberg**
Department of Mathematics
University of Texas at Austin
Austin, TX 78712
`blumberg@math.utexas.edu`

## 1   Stability results for the Multiparameter Persistence Image

The Multiparameter Persistence Image is built from the Vineyard Decomposition associated to the fibered barcode, which is a collection of successive matchings of barcodes. Fix a 2D persistence module $M(f)$. Considering a collection of lines $\{\ell_i\}$, Landi's external stability result [Lan14] implies that as a pair of lines $\ell_i$ and $\ell_j$ get closer together, $\mathrm{bcd}(f_{\ell_i})$ and $\mathrm{bcd}(f_{\ell_j})$ converge. This implies that the persistence image is a sensible construction in the sense that there are not discontinuous jumps in the barcodes along the matchings, provided the cover by the lines is fine enough. Moreover, these considerations also imply that for sufficiently close lines, the vineyard matching closely approximates the bottleneck distance matching.

By Landi's internal stability result [Lan14], for any two 2D persistence modules $M(f)$ and $M(g)$ that are close in the interleaving distance, $\mathrm{bcd}(f_{\ell_i})$ is close to $\mathrm{bcd}(g_{\ell_i})$, with a Lipshitz constant that depends on the slope of the lines. One would hope that this would ultimately imply stability for the multiparameter persistence image. However, stability results will obtain only when these matchings are themselves stable in the 2D persistence module.

The key issue however is that under certain circumstances the trajectories produced by the matchings are arbitrary and hence can be dramatically affected by noise. Specifically, when matching barcodes $\mathrm{bcd}(f_{\ell_i})$ and $\mathrm{bcd}(f_{\ell_{i+1}})$ in which distinct bars $b_{\ell_i}$ and $b'_{\ell_i}$ in $\mathrm{bcd}(f_{\ell_i})$ are matched to a pair of identical bars $b_{\ell_{i+1}}$ and $b'_{\ell_{i+1}}$ in $\mathrm{bcd}(f_{\ell_{i+1}})$ the choice of identification is arbitrary. Composing with a third matching from $\mathrm{bcd}(f_{\ell_{i+1}})$ to $\mathrm{bcd}(f_{\ell_{i+2}})$ which separates the bars $b_{\ell_{i+1}}$ and $b'_{\ell_{i+1}}$ again, both trajectories are possible. Thus, given a path of matchings in which matchings of bars merge, the decomposition into distinct paths of bars cannot be expected to be stable.

We will say that a collection of lines $L = \{\ell_i\}$ is $\kappa$-*generic* for a 2D persistence module $M(f)$ if for any pair of trajectories in the associated Vineyard Decomposition $D_L(f)$ there does not exist an index $i$ such that $\mathrm{m}_i(b_{\ell_i})$ and $\mathrm{m}_i(b'_{\ell_i})$ are closer than $\kappa$ to one another (in the $\| \cdot \|_\infty$ norm between the bars seen as 2D points). Given a $\kappa$-generic trajectory for $M(f)$, if $d_I(M(f), M(g)) < \epsilon < \kappa$, then the paths will not get closer than $\epsilon - \kappa$. As such, we consider stability under the assumption that the perturbations are smaller than $\kappa$. One thing to note is that given the computation of the vineyard decomposition, it is straightforward to compute what the constant of genericity is, which we think of as being akin to a condition number.

We now assume that we are considering trajectories which are $\kappa$-generic and consider stability results for interleaved 2D persistence modules $M(f)$ and $M(g)$ such that $d_I(M(f), M(g)) << \kappa$. (In what follows, we consider the case of a collection of parallel lines for simplicity.) The basis for stability results for the multiparameter persistence image is the following easy geometric result.

**Proposition 1.1.** *Let $\ell_1$ and $\ell_2$ be two adjacent parallel lines parameterized by the vector $\mathbf{e}_\theta$ that are distance $\delta$ apart. Suppose that $b_1 \in \mathrm{bcd}(f_{\ell_1})$ and $b_2 \in \mathrm{bcd}(f_{\ell_2})$ are matched bars in the barcodes along $\ell_1$ and $\ell_2$ respectively. Given another 2D persistence module $M(g)$ such that*

$d_I(M(f), M(g)) < \epsilon$, then the change in the area of the quadrilateral specified by $b_1$ and $b_2$ is bounded by $(2\delta\epsilon)/\tilde{e}$. Along a path of $k$ parallel lines at most $\delta$ apart, the change in area of the region trace out by the bars is bounded by $k \cdot (2\delta\epsilon)/\tilde{e}$.

*Proof.* The change in the parallelogram is bounded by two parallelograms (on the top and on the bottom) with side lengths $\frac{\epsilon}{\tilde{e}}$ and $\delta$. Since the area of a parallelogram is bounded by the area of the rectangle with corresponding side lengths, the bound for a pair of lines follows. For a path of bars, summing along the path yields $(k-1)\frac{\epsilon\delta}{\tilde{e}}$. $\qquad\square$

As an immediate corollary of the proposition, for a collection of lines $\{\ell_i\}$, we can bound the change in the area of the region swept out by the matched barcodes.

**Corollary 1.2.** *Let $M(f)$ and $M(g)$ be 2D persistence diagrams such that $d_I(M(f), M(g)) < \epsilon$ and assume that the sizes of $D_L(f)$ and $D_L(g)$ are bounded by $N$ and each path of matchings has maximum length $c$ before it hits the diagonal. Then*

$$|A_f - A_g| \leq Nc\frac{\epsilon\delta}{\tilde{e}},$$

*where $A_f$ and $A_g$ denote the areas of the regions formed as the union of the parallelograms in $\mathbb{R}^2$ specified by the matchings.*

The multidimensional persistence image is a discretization of this area. To see that it is stable for sufficiently small perturbations, we recall the expression

$$(I_{L,R,p,\sigma}(f))_{i,j} = \sum_{I \in D_L(f)} w(I) \cdot \left(\omega(\ell^*) \exp\left(-\frac{\min_{\ell \in I} \|P_{i,j}, \ell\|^2}{\sigma^2}\right)\right),$$

at each grid point $(i, j)$. Focusing on the Gaussian, If $d_I(M(f), M(g)) < \epsilon$ is sufficiently small, the bar realizing the $\min$ is unchanged and the distance $|P_{i,j}, \ell|^2$ changes by a most $\epsilon$. Since we can bound the change in the value of the Gaussian in terms of $k\epsilon$ (and some additional constants), local stability follows as long as the weightings are themselves stable. For the choices used in the body of the text, this is true by the discussion above and the work of Landi.

## 2 The Multiparameter Persistence Image is finer than the fibered barcode

In this section, we provide a simple example of a pair of graphs that rise to distinct 2D persistence modules that cannot be distinguished with the fibered barcode but can be distinguished by the Multiparameter Persistence Image.

We recall that in the 1D setting, one can compute a persistence diagram when $X$ is a graph $G$ with vertices $V(G)$ and $f$ is a function defined on the nodes of $G$, that is, $f \colon V(G) \to \mathbb{R}$, and is then extended in a piecewise-linear way on the edges of $G$. The topological structures of $G$, such as its branches and loops, can then be encoded in the corresponding persistence diagram.

Figure 1: Each parameter $\alpha$ generates a subgraph of the full graph $G$. A branch pointing downward is detected as a new connected component in the second subgraph, until it gets merged to the other component in the third subgraph. This creates a point in dimension 0 in the corresponding persistence diagram. The loop and the connected component of the graph persist infinitely, leading to points with infinite ordinates.

The following example involves a bifiltration on graphs.

**Example 2.1.** *Let $G_1$ and $G_2$ be two graphs defined with*

$$G_1 = \{\{a, b, c, d\}, \{[a, c], [b, c]\}\}$$

*three connected nodes and one isolated one) and*

$$G_2 = \{\{a, b\}, \{\}\}$$

*(two isolated nodes). Moreover, let*

$$f_1(a) = [0, 1], \; f_1(b) = [1, 0], \; f_1(c) = [1, 1], \; f_1(d) = [1, 1] \text{ and } f_2(a) = [0, 1], \; f_2(b) = [1, 0].$$

*Then $G_1$ and $G_2$ have the exact same fibered barcodes (and thus identical Multiparameter Persistence Landscapes[1] and Kernels) in dimension 0. See Figure 2.*

Figure 2: Vineyard Decomposition, Multiparameter Persistence Image with power $q = 0$, Multiparameter Persistence Image with power $q = 2$ and Multiparameter Persistence Landscape of $G_1$ (top row) and $G_2$ (bottom row). Even though the Multiparameter Persistence Landscapes are the same, both Multiparameter Persistence Images successfully distinguish the graphs (even though it is harder to see for $q = 0$: values are slightly different around the center of the images). Moreover, the Multiparameter Persistence Kernel values (computed with the Sliced Wasserstein Kernel SW [CCO17]) are the same: $\mathcal{K}_{K_{\text{SW}}}(G_1, G_2) = \mathcal{K}_{K_{\text{SW}}}(G_1, G_1) \simeq 138.59$.

## 3  Time series descriptions

Descriptions of sizes and number of classes are available in Table 1.

| Dataset | Train | Test | Length | Nb classes |
|---|---|---|---|---|
| DistalPhalanxOutlineAgeGroup | 400 | 139 | 80 | 3 |
| DistalPhalanxOutlineCorrect | 600 | 276 | 80 | 2 |
| DistalPhalanxTW | 400 | 139 | 80 | 6 |
| ECG200 | 100 | 100 | 96 | 2 |
| GunPoint | 50 | 150 | 150 | 2 |
| ItalyPowerDemand | 67 | 1029 | 24 | 2 |
| MedicalImages | 381 | 760 | 99 | 10 |
| Plane | 105 | 105 | 144 | 7 |
| ProximalPhalanxOutlineAgeGroup | 400 | 205 | 80 | 3 |
| ProximalPhalanxOutlineCorrect | 600 | 291 | 80 | 2 |
| ProximalPhalanxTW | 400 | 205 | 80 | 6 |
| SwedishLeaf | 500 | 625 | 128 | 15 |
| SyntheticControl | 300 | 300 | 60 | 6 |
| GunPointAgeSpan | 135 | 316 | 150 | 2 |
| GunPointMaleVersusFemale | 135 | 316 | 150 | 2 |
| GunPointOldVersusYoung | 136 | 315 | 150 | 2 |
| PowerCons | 180 | 180 | 144 | 2 |

Table 1: Training and testing sizes, and time series length for our experiments.

Computation times for homological dimension 1 are available in Table 2.

Cross-validated resolutions are available in Table 3.

| Dataset | MP-K | MP-L | MP-I |
|---|---|---|---|
| `DistalPhalanxOutlineAgeGroup` | 10109.2 | 631.2 | **135.0** |
| `DistalPhalanxOutlineCorrect` | 53402.7 | 2075.8 | **503.8** |
| `DistalPhalanxTW` | 10251.9 | 354.2 | **80.0** |
| `ProximalPhalanxOutlineAgeGroup` | 13657.6 | 590.8 | **147.2** |
| `ProximalPhalanxOutlineCorrect` | 44267.2 | 1240.0 | **299.0** |
| `ProximalPhalanxTW` | 13913.1 | 238.6 | **56.8** |
| `ECG200` | 1806.0 | 1972.4 | **336.0** |
| `ItalyPowerDemand` | 68895.0 | 1144.0 | **227.9** |
| `MedicalImages` | 74510.8 | 1699.6 | **406.0** |
| `Plane` | 1929.8 | 898.5 | **203.0** |
| `SwedishLeaf` | 22657.8 | 1779.4 | **460.8** |
| `GunPoint` | 2012.0 | 2387.5 | **343.8** |
| `GunPointAgeSpan` | 11829.7 | 2499.1 | **560.0** |
| `GunPointMaleVersusFemale` | 11816.2 | 2580.8 | **628.3** |
| `GunPointOldVersusYoung` | 14067.8 | 3190.4 | **802.7** |
| `PowerCons` | 9002.4 | 2934.9 | **791.7** |
| `SyntheticControl` | 18559.3 | 661.5 | **165.5** |

Table 2: Computation time (s) for time series in dimension 1.

| Dataset | MP-I | MP-L | P-I | P-L |
|---|---|---|---|---|
| `DistalPhalanxOutlineAgeGroup` | 2,500 | 2,500 | 2,500 | 100 |
| `DistalPhalanxOutlineCorrect` | 2,500 | 2,500 | 100 | 100 |
| `DistalPhalanxTW` | 2,500 | 2,500 | 100 | 100 |
| `ProximalPhalanxOutlineAgeGroup` | 2,500 | 100 | 2,500 | 100 |
| `ProximalPhalanxOutlineCorrect` | 100 | 2,500 | 2,500 | 100 |
| `ProximalPhalanxTW` | 2,500 | 100 | 2,500 | 2,500 |
| `ECG200` | 100 | 2,500 | 100 | 2,500 |
| `ItalyPowerDemand` | 2,500 | 100 | 2,500 | 100 |
| `MedicalImages` | 2,500 | 2,500 | 100 | 2,500 |
| `Plane` | 2,500 | 2,500 | 100 | 100 |
| `SwedishLeaf` | 2,500 | 2,500 | 100 | 100 |
| `GunPoint` | 2,500 | 2,500 | 100 | 2,500 |
| `GunPointAgeSpan` | 100 | 100 | 2,500 | 2,500 |
| `GunPointMaleVersusFemale` | 2,500 | 100 | 100 | 100 |
| `GunPointOldVersusYoung` | 2,500 | 2,500 | 100 | 100 |
| `PowerCons` | 2,500 | 2,500 | 2,500 | 100 |
| `SyntheticControl` | 100 | 2,500 | 100 | 2,500 |

Table 3: Best resolutions (selected with cross-validation) for time series.

Cross-validation results are available in Table 4.

## 4 Graph experiments

We made a series of experiments on graph classification. It is natural to imagine that topological features of the graphs might require multiple functions to be fully recovered, see Figure 1. In these experiments, we use the Ricci curvature and the heat kernel signature with time 10 (similarly to what was used in [ZW19, CCI+20]) for computing multiparameter persistence in homological dimension 0. Data come from standard graph classification data sets containing biological and social network graphs. See Table 5 for a description of the graph classification tasks. Moreover, we use the exact same parameters than the ones we used for time series and immunofluorescence images classification (see main paper, Section 4), except for the lines, which are now computed using the minima and maxima of Ricci curvature and heat kernel signatures.

Results are displayed in Table 6. Accuracies are averaged over 5 train/test splits of the data sets obtained with 5 stratified folds. Here, all of the multidimensional persistence summaries have essentially the same performance, although the Multiparameter Persistence Kernel is slightly better

| Dataset | MP-I | MP-L | P-I | P-L |
|---|---|---|---|---|
| DistalPhalanxOutlineAgeGroup | $80.8 \pm 0.17$ | $80.0 \pm 0.14$ | $80.0 \pm 0.15$ | $80.0 \pm 0.13$ |
| DistalPhalanxOutlineCorrect | $80.7 \pm 0.05$ | $76.3 \pm 0.09$ | $74.8 \pm 0.08$ | $71.3 \pm 0.07$ |
| DistalPhalanxTW | $78.0 \pm 0.02$ | $77.0 \pm 0.02$ | $76.8 \pm 0.03$ | $77.2 \pm 0.05$ |
| ProximalPhalanxOutlineAgeGroup | $82.2 \pm 0.11$ | $81.0 \pm 0.13$ | $80.2 \pm 0.11$ | $80.5 \pm 0.12$ |
| ProximalPhalanxOutlineCorrect | $77.5 \pm 0.03$ | $79.3 \pm 0.04$ | $73.7 \pm 0.07$ | $74.0 \pm 0.04$ |
| ProximalPhalanxTW | $77.8 \pm 0.02$ | $78.8 \pm 0.04$ | $76.2 \pm 0.03$ | $75.8 \pm 0.03$ |
| ECG200 | $83.0 \pm 0.07$ | $80.0 \pm 0.12$ | $75.0 \pm 0.11$ | $69.0 \pm 0.06$ |
| ItalyPowerDemand | $86.6 \pm 0.06$ | $86.6 \pm 0.07$ | $71.6 \pm 0.13$ | $59.7 \pm 0.15$ |
| MedicalImages | $60.9 \pm 0.02$ | $57.5 \pm 0.03$ | $55.1 \pm 0.03$ | $54.6 \pm 0.04$ |
| Plane | $89.5 \pm 0.04$ | $89.5 \pm 0.03$ | $88.6 \pm 0.03$ | $79.0 \pm 0.09$ |
| SwedishLeaf | $78.8 \pm 0.04$ | $58.8 \pm 0.01$ | $48.0 \pm 0.04$ | $38.8 \pm 0.03$ |
| GunPoint | $92.0 \pm 0.08$ | $90.0 \pm 0.12$ | $88.0 \pm 0.05$ | $78.0 \pm 0.17$ |
| GunPointAgeSpan | $95.6 \pm 0.02$ | $88.9 \pm 0.04$ | $94.8 \pm 0.02$ | $91.9 \pm 0.06$ |
| GunPointMaleVersusFemale | $97.0 \pm 0.01$ | $89.6 \pm 0.03$ | $89.6 \pm 0.05$ | $85.9 \pm 0.06$ |
| GunPointOldVersusYoung | $99.3 \pm 0.01$ | $95.6 \pm 0.05$ | $97.1 \pm 0.06$ | $95.6 \pm 0.03$ |
| PowerCons | $91.1 \pm 0.1$ | $80.6 \pm 0.06$ | $84.4 \pm 0.06$ | $75.6 \pm 0.09$ |
| SyntheticControl | $56.3 \pm 0.05$ | $53.0 \pm 0.05$ | $50.0 \pm 0.04$ | $44.7 \pm 0.06$ |

Table 4: Cross-validation classification results for time series.

| Dataset | Nb graphs | Nb classes | Av. nodes | Av. edges | Av. $\beta_0$ | Av. $\beta_1$ |
|---|---|---|---|---|---|---|
| BZR | 405 | 2 | 35.75 | 38.36 | 1.0 | 3.61 |
| COX2 | 467 | 2 | 41.22 | 43.45 | 1.0 | 3.22 |
| DHFR | 756 | 2 | 42.43 | 44.54 | 1.0 | 3.12 |
| IMDB-BINARY | 1,000 | 2 | 19.77 | 96.53 | 1.0 | 77.76 |
| IMDB-MULTI | 1,500 | 3 | 13.00 | 65.94 | 1.0 | 53.93 |
| MUTAG | 188 | 2 | 17.93 | 19.79 | 1.0 | 2.86 |
| PROTEINS | 1,113 | 2 | 39.06 | 72.82 | 1.08 | 34.84 |

Table 5: Datasets description. $\beta_0$ (resp. $\beta_1$) stands for the 0th-Betti-number (resp. 1st), that is the number of connected components (resp. cycles) in a graph. In particular, an average $\beta_0 = 1.0$ means that all graph in the dataset are connected, and in this case $\beta_1 = \#\{edges\} - \#\{nodes\}$.

in almost all cases. In this case, the conclusion we draw is that the fibered barcodes alone already contain all the salient topological information. Notice again that the fibered barcodes remain superior to the 1D-persistence summaries, however.

| Dataset | P | MP-K | MP-L | MP-I |
|---|---|---|---|---|
| BZR | $82.7 \pm 2.5$ | $\textbf{86.2} \pm \textbf{2.6}$ | $85.7 \pm 2.5$ | $84.2 \pm 2.3$ |
| COX2 | $76.0 \pm 4.1$ | $\textbf{79.9} \pm \textbf{1.8}$ | $79.0 \pm 3.3$ | $77.9 \pm 2.7$ |
| DHFR | $70.9 \pm 3.1$ | $\textbf{81.7} \pm \textbf{1.9}$ | $79.5 \pm 2.3$ | $80.2 \pm 2.2$ |
| IMDB-BINARY | $54.0 \pm 1.9$ | $68.2 \pm 1.2$ | $\textbf{71.2} \pm \textbf{2.0}$ | $71.1 \pm 2.1$ |
| IMDB-MULTI | $36.3 \pm 1.1$ | $\textbf{46.9} \pm \textbf{2.6}$ | $46.2 \pm 2.3$ | $46.7 \pm 2.7$ |
| MUTAG | $79.2 \pm 7.7$ | $\textbf{86.1} \pm \textbf{5.2}$ | $84.0 \pm 6.8$ | $85.6 \pm 7.3$ |
| PROTEINS | $65.4 \pm 2.7$ | $\textbf{67.5} \pm \textbf{3.1}$ | $65.8 \pm 3.3$ | $67.3 \pm 3.5$ |

Table 6: Classification results for graphs.

## Footnotes

[1]A weighted version of the Multiparameter Persistence Landscape is defined in [Vip20], which could potentially distinguish these two multifiltrations, but it is unclear how to choose the weight and implement it.