[Reviews · NeurIPS 2020]

Review 1

Summary and Contributions: This paper develops a multivariate variant of a persistence image, a feature representation in topological data analysis (TDA). In contrast to previous work in TDA, which is only capable of analysing univariate data (specifically, only scalar-valued filtrations are supported), this work shows how to combine a 'sliced' variant of multi-parameter persistent homology with a fixed-size feature vector to permit the usage of topological features in standard machine learning contexts. Next to providing a brief discussion on the stability properties of the novel method, the paper also analyses the empirical properties and behaviour of multi-parameter persistence images. Experiments are conducted in two very different areas, namely time series classification and image analysis.

Strengths: Since multivariate functions are abundant and TDA has yet to fully encompass this setting, the paper discusses a highly important topic. The main strength of the paper is the smart combination of existing techniques, i.e. sliced or fibred barcodes and vineyards, and the development of a new representation, inspired by persistence images. I expect that this paper will provide new avenues for future research in TDA, and it might inspire more researchers to look into this topic, since many TDA applications are somewhat 'plagued' by univariate filtrations (and at least the bivariate case appears to be rather common in applications).

Weaknesses: Even though I feel favourable about this paper, there are two issues that prevent me from fully endorsing it. First, the paper is lacking clarity in some of the definitions. Second, the experimental section is somewhat problematic. Please see Section 5 for details on clarity. In the following, I will comment on the experimental section. - More details on the DTM calculation are required to make this paper accessible to non-experts. What is the stability of the proposed method with respect to these parameters - When discussing the 'bounding rectangle' in l. 261, does this refer to the maximum and minimum function values of the filtration? - Why are there no standard deviations in Table 1? The text specifies that a cross validation procedure was used (and most of the other tables contain standard deviations), so they should also be shown. - Related to that, the paper should be more clear about the 'baseline' classification accuracy that is to be expected for Table 1. I briefly checked a few data sets, and according to 'timeseriesclassification.com', some values are more than 10% lower than the state of the art. Of course, I do not expect the paper to outperform all existing methods, but in the interest of fairness, details about what constitutes a 'good' result should be added. Having an interest in TDA, I feel strongly about this: applications of TDA should always include other methods as well in order to demonstrate to what extent topological features can be helpful. Additional details on the baselines would strengthen the paper immensely. As a simple suggestion, why not add baseline with Euclidean distance or dynamic time warping distance? While such classifiers are of course not competitive in all cases, they can provide important insights. - Finally, I would be interested in knowing the stability with respect to choosing a set of lines. I understand the theoretical stability results, but I would be interested in how this choice is being performed in practice.

Correctness: All claims in the paper are technically correct. Terminology is used consistently throughout the paper. As far as I can tell, the theoretical claims in the supplementary materials are also correct. There are some minor issues with the definitions. For example, in Definition 3.1, $N$ and $n$ are not used consistently; to my understanding, Figure 2 should use $N$ instead of $n$. Moreover, why does the number of lines differ between the three variants? Would it not also make sense to use $N$ lines with the same slope?

Clarity: The aforementioned lack of clarity is a weakness of the paper. Since I feel favourable about the topic and the paper, I think this lack of clarity could deter non-experts from fully understanding this paper. Here are some suggestions to improve clarity and accessibility: - In the abstract, consider adding details about *why* the multivariate or multiparameter setting is not as well understood; mentioning that no unique representations or decompositions exist here would be useful. - The discussion of shortcomings of persistence diagrams in l. 44-- is somewhat vague. Consider rewriting this and the subsequent sentences. One shortcoming of the 'original space' of persistence diagrams, for example, is the lack of fixed-size representations, making their use cumbersome for machine learning. (moreover, I think the complexity of distance metrics exacerbates this issue; the authors might want to consider mentioning this) - In the Background section, consider adding more explanations about *how* the topological features are being calculated. Is a Vietoris--Rips expansion used, for example? - When introducing multiparameter persistent homology, I would suggest using different terminology than $\leq$ when discussing high-dimensional vectors. The switch between the univariate and multivariate setting should be indicated in the symbols being used; for example, why not use $\mathbf$ to indicate vectors, for example, leaving the normal font for scalars? - I would also define the line $l(t)$ as a function of $t$ instead of a set; currently, it is defined as a set, but used as a function; in particular, the usage in l. 187 is redundant. I would suggest sticking to a function definition. - Moreover, there is a clash of terminology with $l$ in Eq. 2; please consider using a different terminology here. - I would suggest improving the exposition in Definition 3.2; consider providing an example of $D_L(f)$, maybe by showing an illustration or extending Figure 3 accordingly? More specifically, Figure 3 could show the matching or the barcode as an additional inlay. - Consider showing an example of Eq. 2; since the paper draws on the persistence image calculation, it would be helpful to illustrate the calculation process. Figure 4 already describes this to some extent, but in this figure, more details (and axis labels) would be helpful. Since TDA is not a well-known framework (yet!), a more 'gentle' introduction to these concepts would be appreciated by readers. (the use of the matching information could be illustrated directly, for example, since this is one of the unique characteristics of this paper)

Relation to Prior Work: Prior work is adequately referenced for the most part. There are however more machine learning papers from recent conferences that showcase TDA applications (for, among others, graph classification). Consider citing additional papers [this is not required, but for machine learning readers, it might be more relevant to see additional TDA applications in ML, rather than general TDA applications]: - Hofer et al., Deep Learning with Topological Signatures (NeurIPS 2017, https://arxiv.org/abs/1707.040410): to my knowledge, the first publication that combines persistent homology with deep learning, and discusses graph analysis approaches - Kim et al., Efficient Topological Layer based on Persistent Landscapes (https://arxiv.org/abs/2002.02778): a recent preprint on combining persistence landscapes and neural networks - Rieck et al., A Persistent Weisfeiler-Lehman Procedure for Graph Classification (ICML 2019, http://proceedings.mlr.press/v97/rieck19a.html): a TDA-based approach to graph analysis

Reproducibility: Yes

Additional Feedback: --- Update after rebuttal: I thank the authors for their rebuttal and I appreciate the additional clarifications, which I think will help improve the manuscript! The discussions with the other reviewers raised some additional concerns, in particular about additional (persistence-based) baselines, so I am keeping my score as it is. I would urge the authors to emphasise the suggested application of the paper slightly better: as stated in the rebuttal, if the primary target/application is immunofluorescence images, the paper should put a larger focus on this. It would strengthen the submission immensely if the authors could demonstrate that: - the immunofluorescence imaging application requires topological information (in other words, topology-based methods are improving upon existing baselines) - the bi-filtration & the proposed representation are necessary to obtain a certain level of performance I am mentioning this because the computational requirements of persistent homology are already quite high to be begin with, so adding additional complexity needs to be justified by performance, interpretability, or some other benefits. Finally, I want to reiterate that I feel positively about this paper, and I think with additional changes, it has the benefit of making a strong contribution to machine learning! --- Despite my criticisms from above, I feel favourably about this paper and think that, with some improvements, this could be a very strong addition to the TDA literature. Here are some additional suggestions and typos: - 'lower bounds the interleaving distance' --> 'serves as a lower bound for the interleaving distance' - 'with same slope' --> 'with the same slope' - 'in two sense' --> 'in a two-fold sense' - 'Lipshitz' --> 'Lipschitz' - The citation for Landi (2014) seems incomplete - I would suggest not to use citations as nouns. Instead of writing 'See [LW15]' I would suggest writing 'See Lesnick and Wright [LW15]' for more details.


Review 2

Summary and Contributions: This paper introduces "Multiparameter Persistence Images", a summary representation of multiparameter persistent homology. The idea is the compute fibered barcodes from a multi-filtration via restricting the filter to lines. This can be computed via the vineyard algorithm and turned into an image representation with a fixed resolution to be used in downstream ML tasks. Experiments are presented on different datasets where the authors compare against recent multiparameter persistence approaches. The proposed method compares favorably.

Strengths: The premise of the paper, stating the multiparameter persistence offers additional information, is certainly correct and an interesting topic to study in ML problems. Making this TDA approach amenable to ML algorithms is also desirable. While kernel-based techniques have been developed, they typically come with a high computational burden. Hence, studying alternatives, motivated by work on persistence images, seems to be a natural way to go. Technically, the idea of just having to go through the full computational complexity of computing the barcode once and then using the vineyard algorithm is also quite interesting.

Weaknesses: My main concerns with the approach relate to three issues, outlined below: First, while I can agree with the claim of multiparamter persistence offering more information, I do not think that the claim clearly holds up in the experiments. Yes, you can filter, e.g., by distance to the point cloud and density, but you could also do this seperately. While it's a little unclear whether the "Dg" columns (in the experiments) represent exactly this, even if they would, there are more discrimiative and machine-learning compatible approaches to do this, e.g., work on learnable representations by Carriere et al., or Hofer et al., which are simpler to compute and typically offer better performance on most tasks. Comparing against existing multi-parameter persistence work is totally fine, but the authors should have included stronger baselines. Some questions that immediately came up are: What was the resolution for the multiparameter persistence images? As far as I understood, its 10x10 or 50x50 (page 7), but as it is written it's somewhat unclear. If the resolution would be, say 50x50, the XGBoost classifier is operating in a 2500-dim. space, as you need to vectorize first and so, performance improvements could potentially be explained by that? If I am misunderstanding this, please clarify in the rebuttal. Second, I do think the claim that the approach will work with actual MULTI-parameter persistence (i.e., > 2) is quite bold. E.g., it's not clear to me how the number of lines should be chosen if much more parameters would be available. I would assume the lines would also have to be "close" in order for all the theory to hold. In a setting with more parameters, computing single-parameter persistence and, e.g., concatenating vectorized versions of the barcodes, seems much more accessible and computationally feasible. I also think that the discussion on the subtleties of the stability of the matchings (following Prop. 3.4.) is weak. While there is the claim that instabilities can be detected (or they disappear for well-bahaved multi-filtrations) algorithmically, the reader is left with little insight whether this is an actual problem or not. I do find the stability result interesting, but without further insights, the result seems a little disconnected. Finally, and importantly for the community, I found the notation throughout the paper to be quite hard to digest. E.g., what is "n" in Def. 3.3.? My guess is that this is a typo and should be p, right? (as the image has resolution p x p). I would also guess that the functions need to be bounded. Also, in the same definition, what is a,b,c,d - this can not be totally arbitrary, right? Further, some confusion also came up with the notation bcd(f|_l), particularly the f|_l part, denoting the restriction of f to l: I mean, f is defined on X, so how does this restriction work - there seems to be something missing here ...

Correctness: Technically, the paper appears correct, modulo some typos, but the notation is sometimes quite confusing. Also, some of the claims do not hold up empiricially, in my point of view (see detailed comments).

Clarity: Overall, the reader can follow the key ideas, however, when going into more detail, the paper is very hard to follow and would clearly benefit from additional editing and rounding off the edges. I think, in it's current form, the NeurIPS community would have a hard time following the presented material.

Relation to Prior Work: Prior work on multi-parameter persistence is well discussed and the presented approach also differs to these works in important aspects, but the advantages of multi-parameter persistence over existing (much simpler) single-parameter persistence) approaches are not clearly worked out. I think it's totally fine to go through more (computational) trouble in terms of constructing the multiparameter-persistence images, but there should also be a clear, demonstratable gain.

Reproducibility: Yes

Additional Feedback:


Review 3

Summary and Contributions: I thank the authors for their response. After discussions, while I am positive about the potential of this paper, I agree with other reviewers that the paper needs stronger baseline for comparisons, including comparing with state-of-the-art 1D persistence based approaches. Topological data analysis has attracted much attention lately. In particular, the use of persistent homology based summaries for different types of data has found successful applications in different practical domains. This paper develops a topological summary, called multiparameter persistence images, for multi-parameter (mostly 2D) persistence, and shows that it performs on par or better than other topological summaries for the multi-parameter case. Multi-parameter settings are very important in practice, and having an effective, concise, and easily commutable topological summary for such setting is highly desirable. However, multi-parameter persistence modules do not admit simple decomposition (complete discrete representations) as is the case for 1D persistence. Previous attempts for computing a summary for multi-parameter case typically boil down to taking ``slices'' (i.e, restricted it to lines) and computing persistence diagrams for those 1D slices. The approach proposed in this paper still relies on these 1D slices. However, the key innovation is that it also captures the ``connection" between these 1D slices via the vineyard algorithm. This makes the new topological summary more informative than the previous ones based only persistence diagrams for individual 1D slices. Detailed strength/weakness are below. Overall I am quite positive about this paper, mainly due to that I think multi-parameter persistence based summary is an important topic. While what's developed in this paper can perhaps be improved, it is more informative than previous approaches and the python package will be made available, which will help facilitate the use of it in practical applications. Current experimental results also show the effectiveness of this new summary.

Strengths: + As mentioned above, developing an effective, concise, and easily commutable topological summary for the multi-parameter setting is highly desirable, and will help further broaden the scope of topological data analysis. This paper makes a good attempt for addressing this. + The proposed multi-parameter persistence images encodes more information than previous approaches. Experimental results also demonstrate this. + The proposed multi-parameter persistence image has a finite vector representation. Thus it will be easy to combine this representation with existing ML pipelines, including neural networks. + The python package is developed and will be made public available. The availability of easy to use tools for topological data analysis is crucial in facilitating the applications of these methods.

Weaknesses: - Existing experiments are quite convincing. But it will be stronger if there had been more baselines, such as more methods based on just single parameter persistence, and/or other simple summarization of multiple persistence diagrams computed from 1D slices. (A minor point: it will be useful to report the time it takes to compute these summaries.) - There is a concern over the stability of the matching computed by vineyard (the theoretical results do not direclty apply to it in practical setting). I don't expect this to be addressed theoretically, but perhaps some empirical studies can be carried out to show that in practice, these summaries are rather stable.

Correctness: I don't detect any issue from the main text. I read the Supplement as well and it also looks good.

Clarity: The paper is overall very well-written. I have only two comments: -- can you elaborate on the matching computed from the vineyard algorithm? (This can be put in Supplement.) This is important especially given that the stability of the approach depends on this. -- Can you elaborate the example in Figure 4: (1) what line set did you use to generate that multi-parameter persistence image? (2) why is that group of long bars necessarily indicate one major circle in data? how should one interpret that? How should one say differentiate the case behind there is one major loop versus multiple loops?

Relation to Prior Work: Adequate

Reproducibility: Yes

Additional Feedback:

[Author Response · NeurIPS 2020]

We thank the reviewers for their careful and thoughtful reviews.

**Concerns about the time series experiments (R1, R2, R3)**

The reviewers pointed out that we did not use standard baselines; we agree the paper would be
strengthened by including them. Hence, we compared with Euclidean nearest neighbor (B1), dynamic
time warping with optimized (B2) and constant window width (B3), persistence scale-space kernel
on the Rips filtration (without density), (R), standard 1D persistence image (I) and landscapes (L) of
a weighted average of Rips and density, and the multiparameter kernel (MP-K), landscape (MP-L),
and image (MP-I). We show results for only a few datasets due to lack of space.

| Dataset | B1 | B2 | B3 | R | I | L | MP-K | MP-L | MP-I |
|---|---|---|---|---|---|---|---|---|---|
| GunPointOldVersusYoung | 95.2 | **96.5** | 83.8 | - | 98.7 | 95.9 | 99.0 | 97.1 | **100.0** |
| ProximalPhalanxOutlineCorrect | **80.8** | 79.0 | 78.4 | 78.4 | 67.4 | 72.5 | 78.7 | 78.7 | **81.8** |
| ECG200 | **88.0** | **88.0** | 77.0 | 67.0 | 68.0 | 68.0 | 77.0 | 74.0 | **83.0** |
| Plane | 96.2 | **100.0** | **100.0** | 82.9 | 64.8 | 82.9 | 92.4 | 84.8 | **97.1** |
| SyntheticControl | 88.0 | 98.3 | **99.3** | 50.0 | 45.7 | 44.0 | 50.7 | **60.3** | 56.3 |

8
We interpret this table to indicate that multiparameter persistence is a competitive option for this
problem, and our method is the best of the persistence-based approaches. However, our view is that
the real virtue of this method comes in situations such as the immunofluorescence images experiment.

**Standard deviations** (std) on the train set for these results when cross-validating were on the order
of 0.05. There is no std for the test set since we used suggested train / test splits. **Computation**
**times** for MP-I on these tasks averaged around 350 seconds; this was 3-4 times faster than MP-L and
around 30-50 times faster than MP-K. The other summaries based on 1D persistence had running
times similar to MP-I. **Size of feature space** It is correct that we worked with 10x10 and 50x50
images. We cross-validated all of the methods with up to 2,500 features, to ensure fair comparison;
hence we do not think the size of the feature space accounts for performance improvements.

**Stability with respect to matching (R1, R2, R3)**

We agree with the reviewers that there is a disconnect between the theory and stability in practice.
We will expand the discussion of this in the revision.

**Perturbing the lines** We experimented with random perturbations of the lines selected (both by
randomly shifting the angles and making them closer and further apart). Accuracies degraded slowly
in the size of the perturbations. For instance, when perturbing the line endpoints with noise whose
amplitude was up to 10 times the distance between the lines, the mean accuracy of ECG200 and
GunPoint went down by only 2-3%, with std around 0.02. **Matching instability** We never detected
this in practice in any of the examples we ran, although it is easy to create synthetic examples
exhibiting it. We will detail how to detect it (by comparing with bottleneck matchings) in the revision.

**Extension to multiparameter persistence (R2)**

We agree with R2 that extending the MP-I to multiparameter persistence is potentially subtle, and
requires discussion of the choice of lines. For example, in $R^3$, one would sweep the 3D space
with planes that all intersect on a given line. Since each plane would itself be swept with lines, 2D
summands can be computed in each plane, and then connected through the common line to generate
3D summands. The stability result generalizes to this situation with a different geometric base case.

**Clarity (R1, R2, R3)**

All reviewers made very useful comments about how to improve clarity of the paper, which we
will incorporate. We also thank R1 for the writing comments and reference works. We answer
their questions here. **DTM**: we didn't study stability w.r.t. the DTM parameters but this is a very
interesting question to look at. **Bounding rectangle** refers to the rectangle given by the minima and
maxima of both filtrations. **Fig 2**: $n$ should be replaced by $N$. **Lines**: The reason why $N$ is not fixed
for lines with same slope is because we implemented it that way. There is no theoretical obstruction
for this though. $f|_\ell$ is indeed an abuse of notation. **Def 3.3**: $n$ should be $p$ and $a, b, c, d$ should satisfy
$a \leq b, c \leq d$. **Fig 4** was generated with lines of same slope. Sets of large bars indicate topological
features that are persistent in both filtrations: in our case, circles with large diameters that are made
of points with large density. Smaller loops would generate bars with smaller lengths for at least one
filtration. Difference between a big loop and many small loops shows in the decomposition, but can
be missed in MP-I since all the Gaussian functions (corresponding to the sets of bars) are summed.



[Meta-Review · NeurIPS 2020]

The paper, the reviews, the author response and the ensuing discussion were all taken into consideration. Two of three reviewers considered the work marginally above the acceptance threshold and one considered it marginally below the threshold. Concerns, after taking the author response into account, included missing (stronger) baselines, stability in practice, and claims about working with multiparameter persistence and it offering more information. On the other hand, the topic and smart aspects of the technical solution were considered interesting, and able to inspire future research. Overall the paper may be of sufficient quality to be presented at NeurIPS.